# Rural-Urban Disparities in Vaccine Hesitancy among Adults in South Tyrol, Italy

**DOI:** 10.3390/vaccines10111870

**Published:** 2022-11-05

**Authors:** Verena Barbieri, Christian J. Wiedermann, Stefano Lombardo, Barbara Plagg, Timon Gärtner, Dietmar Ausserhofer, Wolfgang Wiedermann, Adolf Engl, Giuliano Piccoliori

**Affiliations:** 1Institute of General Practice and Public Health, Claudiana—College of Health Professions, 39100 Bolzano, Italy; 2Department of Public Health, Medical Decision Making and Health Technology Assessment, University of Health Sciences, Medical Informatics and Technology, 6060 Hall, Austria; 3Provincial Institute for Statistics of the Autonomous Province of Bolzano—South Tyrol (ASTAT), 39100 Bolzano, Italy; 4Faculty of Education, Free University of Bolzano, 39100 Bolzano, Italy; 5Department of Educational, School and Counseling Psychology, Missouri Prevention Science Institute, College of Education and Human Development, University of Missouri, Columbia, MO 65211, USA

**Keywords:** COVID-19, vaccination, hesitancy, rurality, primary care

## Abstract

Background: The demographic determinants of hesitancy in Coronavirus Disease—2019 (COVID-19) vaccination include rurality, particularly in low- and middle-income countries. In the second year of the pandemic, in South Tyrol, Italy, 15.6 percent of a representative adult sample reported hesitancy. Individual factors responsible for greater vaccination hesitancy in rural areas of central Europe are poorly understood. Methods: A cross-sectional survey on a probability-based sample of South Tyrol residents in March 2021 was analyzed. The questionnaire collected information on sociodemographic characteristics, comorbidities, COVID-19-related experiences, conspiracy thinking, and the likelihood of accepting the national vaccination plan. A logistic regression analysis was performed. Results: Among 1426 survey participants, 17.6% of the rural sample (*n* = 145/824) reported hesitancy with COVID-19 vaccination versus 12.8% (*n* = 77/602) in urban residents (*p* = 0.013). Rural residents were less likely to have post-secondary education, lived more frequently in households with children under six years of age, and their economic situation was worse than before the pandemic. Chronic diseases and deaths due to COVID-19 among close relatives were less frequently reported, and trust in pandemic management by national public health institutions was lower, as was trust in local authorities, civil protection, and local health services. Logistic regression models confirmed the most well-known predictors of hesitancy in both urban and rural populations; overall, residency was not an independent predictor. Conclusion: Several predictors of COVID-19 vaccine hesitancy were more prevalent in rural areas than in urban areas, which may explain the lower vaccine uptake in rural areas. Rurality is not a determinant of vaccine hesitancy in the economically well-developed North of Italy.

## 1. Introduction

Infectious diseases are part of human life and, as in the case of Severe Acute Respiratory Syndrome Coronavirus-2 (SARS-CoV-2) with its pandemic spread, can pose a global threat to humanity. In addition to climate change, natural resource depletion, and wars, Coronavirus Disease-2019 (COVID-19) adversely affects personal well-being and mental health [1]. Although vaccines have been developed as a control mechanism against SARS-CoV-2, vaccine acceptance depends on ideas, sentiments, or information. Despite the efficacy and safety of COVID-19 vaccines, acceptance has been heterogeneous and remains low depending on countries and regions, and current rates in western countries often fall short, particularly for the required COVID-19 booster vaccinations. Social determinants likely play an important role in vaccine acceptance, and not only the biological system of infection but also the social system concerning the disease is contagious [2].

Successful prevention requires not only safe and effective vaccines but also an understanding of the factors that determine the willingness to be vaccinated and how best to communicate information leading to vaccination and boosters [3]. It is critical to better understand what determines vaccine uptake, including vaccination intention and the factors that could determine hesitancy and resistance. The determining factors of hesitancy include individual and social characteristics, such as distrust in authority, risk aversion or disgust sensitivity, age, economic situation, educational status, and type of parenthood [4]. 

South Tyrol (with a total population of approximately 525.000) is the alpine part of Trentino—Alto Adige, a northwestern autonomous region in Italy. At the end of 2021, the weekly incidence rates of the COVID-19 pandemic made South Tyrol among the worst of all Italian regions, and its vaccination rate was the lowest in Italy, although its quality of public healthcare is above the Italian average [5]. In a representative survey performed in March 2021, 15.6 percent of the sample reported vaccine hesitancy; increased hesitancy was mostly observed in young ages, the absence of chronic disease, a worsened economic situation, mistrust in institutions, and conspiracy thinking [6].

On 16 December 2021, the New York Times related the pandemic situation in that region to the suggestion, among others, of rurality and historical mistrust in South Tyrol against the central Italian government [7]. Unsurprisingly, in the March 2021 survey of South Tyrol, vaccine hesitancy was significantly higher in the rural than in the urban population [6]. Rural populations, along with vulnerable and excluded people, are among those for whom improved vaccination rates and access to care are urgently needed for the prevention and treatment of COVID-19 [8]. Prior to COVID-19, rural communities have been shown to have lower vaccination rates for influenza and pneumonia than urban communities [9,10]. A systematic review of vaccine hesitancy identified preferentially in studies from the United States of America factors including parental hesitancy, negative clinic experiences, referrals outside of primary care settings, and distance to providers as barriers to vaccination in rural settings [11]. Low vaccination rates have been observed, particularly in low- and middle-income countries [8], and lower vaccination rates in rural populations have continued to worsen during the COVID-19 pandemic [12,13,14]. COVID-19 vaccination intention was not affected by rural or urban residence in a large international study from Australia, Canada, England, New Zealand, and the United States [15], and a study from West Africa [16]; however, the majority of surveys reported increased vaccine hesitancy in rural people [17,18,19,20,21,22].

In a probability-based survey, greater vaccine hesitancy was previously reported in rural residents of South Tyrol than in urban residents [6]. With no data on factors associated with rural vaccine hesitancy included in that report, it is crucial to estimate its determinants; therefore, interventions may be implemented if necessary. Since factors of hesitancy in rural South Tyrol have not yet been reported, we re-examined the survey results for rural-urban disparities by comparing the individual and social characteristics of vaccine hesitancy in the two populations.

## 2. Materials and Methods

### 2.1. Study Design and Data Collection

The study design and data collection have been reported in detail elsewhere [6]. As described previously, this study used a probability-based mode survey. Participants were aged >18 years. The survey was conducted in March 2021. To obtain a high level of congruence between the distribution of demographics in the sample and the adult population (regarding age, gender, and living area), a sample size of *n* = 1000 was recommended by the World Health Organization Regional Office for Europe, who developed the survey [23]. A sample size of 1000 is recommended for surveys with large populations. as the precision of the survey only increases little [24]. More than 4000 of the 430,000 full-aged South Tyrolean inhabitants were invited to participate, following a one-stage random sampling design stratified for the quantitative study. The sample size was defined based on an expected participation rate of 33%, as observed in previous surveys [25]. Participants were invited via letter, including the planned participation date; a link to the online questionnaire (with telephone support) covering demographic, clinical, and socio-behavioral aspects and a personalized password for use as a pseudo-anonymization code.

### 2.2. Rurality

Rurality was measured in the context of local circumstances. South Tyrol is an alpine region with approximately 533.000 inhabitants and an area of 7.400 km^2^. According to the official definition, we defined eight towns (219,340 inhabitants) as urban areas and all other villages as rural areas.

### 2.3. Questionnaire

As previously described [6], the questionnaire was an extended version of the COSMO survey [26]. Questions regarding hesitation to be vaccinated were also included. The intelligibility and validity of the questionnaire were discussed in a published survey tool [23]. For the assessment of a health literacy-related construct, items were adapted from Sørensen et al. [27]. For the self-assessed probability and susceptibility of contracting COVID-19, validated items were adapted from Brewer et al. [28]. COSMO contains instruments to measure trust in sources of information and institutions [29,30], conspiracy perceptions [31], resilience [32], and altruism [33]. Sociodemographic questions were adapted to the South Tyrolean context, including items for the municipality and mother tongues of German, Italian, and Ladin. In addition, specific questions regarding social behavior and well-being have been added [34].

### 2.4. Vaccine Hesitancy (Dependent Variable)

Vaccine hesitancy was measured using the dichotomous question, ‘Would you be vaccinated against COVID-19?’. For the subgroup of parents of 0–6 years old children the question ‘Would you have your child vaccinated?’. To obtain more detailed information, questions about trust in COVID-19 vaccination during the pandemic and non-COVID-19 vaccination of children, beliefs about the COVID-19 vaccine itself and the vaccines in the national vaccination plan, and opinions regarding COVID-19 vaccination and non-COVID-19 vaccination for children, in general, were added.

### 2.5. Putative Predictors of Vaccine Hesitancy (Independent Variables)

Sociodemographic variables including age, gender, mother tongue, urban/rural residence, educational status, citizenship, information about the living situation, healthcare profession, chronic disease, and economic situation in the last three months were assessed as previously described [6]. Predictors for COVID-19 vaccination were obtained from literature and the COSMO questionnaire. Trust in information sources and institutions (health authorities and politics), resilience, well-being within the last two weeks, and altruism were assessed using validated instruments [6].

### 2.6. Statistical Analysis

Metric data are presented as medians (first quartile, third quartile) due to non-normally distributed variables, and significant differences between the two groups were calculated using the Mann–Whitney U-test. Nominal and ordinal data are presented as absolute numbers and percentages. The chi-square test was used to test for differences and correlations.

The sum scores were calculated for conspiracy theories (Cronbach’s alpha = 0.811; 95% CI = [0.795; 0.826]), well-being (Cronbach’s alpha = 0.876 [865; 886]), resilience (Cronbach’s alpha = 0.664 [0.632; 0.693]), altruism (Cronbach’s alpha = 0.795 [0.778, 0.812]), trust in the media (Cronbach’s alpha = 0.844 [0.830; 0.856]), and trust in institutions (Cronbach’s alpha = 0.921 [0.915, 0.927]). Details of the questionnaires used are described in [21].

For trust in institutions and in the media, the response category ‘I do not know’ was coded as 3.5, which corresponds to the median value of the Likert scale. Details can be found in [6]. A higher sum score indicates greater trust. 

For comparisons of the sum scores for more than two groups, we used the Kruskal–Wallis test with posthoc tests for pairwise comparisons. Posthoc Mann–Whitney tests were performed with Bonferroni correction for multiple tests.

Logistic regression was used to explain vaccine hesitancy according to the predictor variables for urban and rural residents. Metric variables were checked for the linearity of the logits by testing the quadratic term for significance. Model diagnostics were performed using DFBETA statistics. The minimum sample size for a logistic regression model of 550 was calculated as previously described [6].

*P*–values < 0.001 are indicated with ***, <0.01 with **, <0.05, *, and *p*–values ≥ 0.05, are regarded as not significant (n.s.). All statistical analyses were performed using the SPSS version 27.

## 3. Results

### 3.1. Population Characteristics

Data from 1426 individuals were collected. The demographic characteristics of the study sample were representative of age, sex, municipality, and native language. Comparisons of demographic variables with respect to vaccine hesitancy have been previously presented [6]. Table 1 compares the sample characteristics of urban and rural residents. Rural residents were less likely to have a university degree but were more likely to have an education in vocational schools. Rural residents reported German or Ladin as their mother tongue significantly more frequently than Italian or other minority languages. They lived more frequently in households with children under six years of age, their economic situation was worse than before the pandemic, and chronic diseases and deaths from COVID-19 among close relatives or friends were less frequently reported. Age, gender, Italian citizenship, all other household structures, work in health professions, and COVID-19 infection rates were not significantly different between the rural and urban populations.

A total of 17.6% (*n* = 145) of the rural sample reported COVID-19 vaccine hesitancy versus 12.8% (*n* = 77) in the urban resident group (*p* = 0.013). Regarding general non-COVID-19 vaccination, no significant differences were found in parental hesitancy between the urban and rural populations.

Significant differences between hesitant and non-hesitant participants (details in [21]) were found for age (older people were less hesitant), educational level (participants with an educational level of university were less hesitant), and language (participants speaking more than one or other languages were more hesitant). Participants from households with children aged 0–6 years were significantly more hesitant, whereas those from households with at-risk COVID-19 patients were significantly less hesitant. Participants with chronic diseases and higher economic status were less hesitant. In a parental study on non-COVID-19 vaccination of children, participants who were more parental non-COVID-19 vaccine hesitant were even more hesitant regarding their COVID-19 vaccination.

### 3.2. Vaccination Perception

#### 3.2.1. Compulsory Vaccination for Non-Coronaviruses

Comparing the baseline characteristics of differences between urban and rural residents in the hesitant and non-hesitant groups separately (Appendix A), significant differences were found in both groups for educational level and native language. In both groups, urban residents had university degrees more frequently (non-hesitant: 28.8% vs. 15.7%; *p* < 0.001 and hesitant: 21.1% vs. 9.0%, *p* = 0.03). Rural areas were mostly populated by Germans (non-hesitant: 80.7% and hesitant: 77.2%), whereas urban areas were mostly populated by Italians (non-hesitant: 53.1% and 45.5%). The overall *p*–value for both groups was <0.001. Age was significantly different between urban and rural residents, but only in the hesitant group (*p* = 0.001). Younger, hesitant people were found more often in rural areas (18–34 years: 43.8%) than in urban (18–34 years: 28.9%). Only the group of hesitant participants suffering from a chronic disease was found significantly more often in urban residents (10.5%) than in rural residents (3.4%, *p* = 0.034). A significant difference in the economic situation was detected only in the non-hesitant group (rural: 26.8% worse vs. urban: 19.7%; *p* = 0.02).

Comparing the baseline characteristics for differences between hesitant and non-hesitant participants separately for urban and rural residents (Appendix A), significant differences were found in both groups with respect to age (hesitant persons were significantly younger than non-hesitant, urban: *p* = 0.005; rural: *p* < 0.001), living with COVID-19 patients at risk, suffering from a chronic disease, and economic situation. In detail, in the urban areas, 22.9% of the non-hesitant lived together with a patient at risk, while only 9.1% of the hesitant did (*p* = 0.006). In rural areas, 22.8% were non-hesitant and 13.8% were hesitant (*p* = 0.016). In the urban areas, 21.8% of the non-hesitant and 10.5% of the hesitant suffered from a chronic disease (*p* = 0.023) and in the urban areas, we found 17.5% of the non-hesitant and 3.4% of the hesitant (*p* < 0.001) to suffer from a chronic disease. Significantly more hesitant participants had worse economic situations than non-hesitant participants (urban: 34.2% vs. 19.7%; rural: 43.8% vs. 26.8%). Furthermore, in urban areas, we found a significant difference between non-hesitant and hesitant participants for the factor “working in the health care sector” (7.4% vs. 1.3%, *p* = 0.046). In rural areas, the educational status differed significantly between non-hesitant (15.7% with a university degree) and hesitant (9.0%) participants (*p* = 0.004). Even in the group of parents of children from 0–6, we found a significant difference between non-hesitant (12.4% had children) and hesitant (22.1% had children (*p* = 0.002) in the rural area), as well as in the single group (17.8% of the non-hesitant were single and 11.0% of the hesitant, respectively, *p* = 0.046).

Participants with children aged up to six years (*n* = 178) were asked whether the pandemic changed their attitudes toward compulsory vaccination for non-coronaviruses (Appendix A). Almost three-quarters (73.8%) of the participants did not change their attitude toward compulsory vaccination of children against non-coronaviruses, one-fifth (19.6%) changed their attitude toward higher support because of the pandemic, and 6.6% supported it less now. This change was not significantly different between participants in the urban (74.5% no change, 19.6% more support) and rural (73.3% no change, 19.5% more support) areas.

In general, differences between rural and urban populations regarding compulsory vaccination are sparse. Agreements with political decisions to increase compulsory vaccination ranged from 54.7% to 57.9%. Significantly more urban (46.8%) than rural (27.6%) residents believed that the consequences on their children’s health would be serious without compulsory vaccination.

Regarding the necessity for compulsory vaccination, no significant differences were observed (67.2% of rural and 74.2% of urban residents (rather) agreed that it was efficient (12.1% and 6.3%, respectively); agreed that the natural immune system was completely sufficient). Of the rural and urban residents, 3.4% and 4.8% (rather) agreed that diseases for vaccination no longer exist, and 16.4% and 7.9% (rather) agreed that the whole endeavor is just a profit for the pharmaceutical industry, respectively.

Regarding belief in harmfulness, no significant differences between rural and urban respondents were detected: 11.3% of the rural and 4.8% of the urban population (rather) agreed that the risks were greater than the protection and 13.8% and 4.8% (rather) agreed that vaccines were not sufficiently controlled. Rural residents (7.8%) and urban residents (6.3) rarely (rather) agree that there are doctors who advise against it, while 6.9% and 3.2% (rather) agree that there have been negative experiences with vaccination in the family, respectively.

Overall, even if statistically insignificant, questions regarding the necessity and harmfulness of compulsory vaccinations were answered slightly more confidently by the urban residents. The position of compulsory vaccination was not significantly different as assessed by the urban and rural populations; 67% of the rural and 81% of the urban residents (rather) agreed that it is important that a child receives the protection it needs, and 55.2% and 66.7% (rather) agreed that it is important to guarantee that 17.2% of the rural and 21% of the urban population (rather) agreed that access to compulsory vaccination is difficult at the moment, while 26.7% and 25.8% are (rather) concerned about the decline in compulsory vaccination due to the pandemic, respectively; 38.8% of the rural and 45.2% of the urban residents (rather) agreed that it would be more reassuring to vaccinate the child at the doctor’s office, while 14.7% and 14.3% (rather) agreed that compulsory vaccination in the current situation could wait.

For all questions regarding the importance of compulsory vaccination during the current pandemic, although not significantly different, the urban population was more convinced of compulsory vaccination than the rural population. The most important reasons for mandatory vaccination of children were protection (69.8% of rural participants and 82.3% of urban participants) and responsibility (58.6% and 61.3%, respectively). Sanctions (6.3%) and the expectations of family and friends (2.0%) were rarely mentioned as reasons for compulsory vaccination. The most important reason for hesitation in non-COVID-19 vaccination was the possibility of infection or the side effects of vaccination (10.6% rural vs. 5% urban).

#### 3.2.2. Vaccination for Coronaviruses

The attributes of the sample regarding COVID-19 vaccination and comparisons between urban and rural residents are given in Appendix A. Urban residents significantly (*p* < 0.001) more often (rather) agreed that everybody should be vaccinated according to the national plan (69.7%) than rural residents (58.5%); 33% of rural and 40.5% of urban residents expected (rather) serious consequences for their daily lives if they did not receive the COVID-19 vaccination (*p* = 0.001).

Of the five statements listed on the COVID-19 vaccine, only the question ‘If I knew that I was already infected with COVID-19, I would not get the vaccine’ was significantly (*p* = 0.007) different between rural and urban (29.3% vs. 23.6% (rather) agreed, respectively); 80.6% of urban and 77.3% of the rural residents believed that vaccination could help contain the spread of the COVID-19 virus, 11.0% of the rural and 9.3% of the urban residents think that they should not get vaccinated if everybody else is vaccinated, and 73.6% of the rural population and 77.4% of the urban population (rather) agreed to get vaccinated if the doctor recommended the vaccination.

The perceived necessity of the COVID-19 vaccination was measured using four questions. While vaccination was regarded as (rather) effective by 70.7% of the rural population and 75.4% of the urban population, with no significant difference, herd immunity was significantly (*p* < 0.001) regarded as (rather) insufficient by the urban (71.6%) compared to the rural (62.3%) residence group. Although 77.8% of rural residents did (rather) not agree that the disease was trivial or did not exist, urban residents expressed this opinion significantly more often (84.7%, *p* = 0.003). Rural residents (15.4%) strongly (rather) agreed that the whole thing is just a profit for the pharmaceutical industry compared to urban residents (9.5%) (*p* = 0.003).

The harmfulness of COVID-19 vaccination was assessed using four questions. Three of them were significantly different between urban and rural residents. Additional risks in the RNA of the vaccine were seen as (rather) not harmful to the urban population (55.9% vs. 48.7%; *p* = 0.006), and doctors who advised against it were seen more often as (rather) harmful in the urban population (64.1% vs. 58.4%, *p* = 0.037), while the reason ‘an obligation to vaccinate certain groups with priority will lead to great socio-political discussion’ was answered as neutral rather than the urban population (39.7% vs. 30.6%, respectively, *p* = 0.002). Unknown long-term risks were regarded as (rather) harmful by 33.1% of the population, with no significant difference between rural and urban residents. 

The reasons for vaccination against COVID-19 were the responsibility of citizens (64.8%) and self-protection (58.8%), whereas sanctions (1.6%) and expectations of family and friends (8.4%) were rarely mentioned. Responsibility of citizens was reported significantly more often as an important reason for urban residents (69.9%) than for rural residents (61.0%) (*p* = 0.001). Reasons for COVID-19 vaccination were mentioned sparsely; the most important were fear of being infected or other side effects (10.2%) and mistrust in the effectiveness of the vaccine (7.9%).

#### 3.2.3. Vaccine-Hesitancy-Related Attitudes in Residence Groups

In the non-hesitant group, answers to questions regarding vaccine hesitancy-related attitudes concerning decision-making, trust in COVID-19 vaccination, and the necessity and harmfulness of COVID-19 vaccination were consistent (*p* < 0.001). All of these questions were investigated for differences between the urban and rural participants for the total sample, as well as for the subsample of parents of children aged–0–6 years. 

Agreement with the decision–making regarding COVID-19 (overall agreement 57.8%), the COVID-19 vaccination plan (75%), and the general national vaccination plan (73.2%) were not significantly different between the urban and rural populations. In the subgroup of parents of young children, we found a significant difference only in agreement with decisions regarding COVID-19 vaccination (82.1% urban vs. 67.3% rural, *p* = 0.044). Decisions regarding COVID-19 strategies and the general national vaccination plan were not significantly different, and participants (rather) agreed with 52.7% and 67.5%, respectively.

### 3.3. Predictors of Vaccine Hesitancy

#### 3.3.1. Vaccine-Hesitancy-Related Attitudes in Residence Groups

In our previous report [6] the significant predictors of vaccine hesitancy were lower age, lower educational level, non-suffering from chronic disease, lower frequency of information search, higher disagreement with the national vaccination plan, lower trust in institutions, and higher belief in conspiracy theories.

Educational level, chronic diseases, economic status, frequency of information searches, agreement with the national vaccination plan, and trust in institutions also differed significantly between urban and rural residences (Appendix A).

Age was a significant predictor of vaccination hesitancy but was not significantly different according to residence. Furthermore, even if not significant in the regression model, significant differences between hesitant and non-hesitant participants were found for persons living with COVID-19 patients at risk or with children aged–0–6, persons with a worse economic situation within the last three months, trust in media, and persons having already been infected with the COVID-19 virus. Of these variables, only living situations with children aged 0–6 years were significantly different between urban and rural participants.

Finally, we found that altruism, resilience, well-being, and close relatives or friends who died from COVID-19 were significantly different between urban and rural residents but not between hesitant and non-hesitant participants.

#### 3.3.2. Attitudes of Rural Vaccine Hesitancy

Trust in information sources and institutions—Overall (*n* = 1426), the frequency of searching for information on COVID-19 vaccination was significantly different between urban (53.2% searched more than once a week) and rural (37.4%) residents (*p* < 0.001).

Furthermore, trust in the Ministry of Health was significantly higher among urban residents (65.9% vs. 53.3%, *p* < 0.001), as was trust in the National Institute of Health (62.8% vs. 42.8%, *p* < 0.001), regional toll-free and emergency numbers (51.7% vs. 37.0%, *p* < 0.001), civil protection (69.4% vs. 63.3%, *p* = 0.007), provincial government (56.4% vs. 48.8%, *p* = 0.012), and management of the South Tyrolean Health Service (66.3% vs. 51.4%). Overall, trust in healthcare workers was approximately 70.8% and trust in the WHO was approximately 57.7%. Both were not significantly different between the urban and rural participants.

In the subpopulation of parents of children aged 0–6, significant differences between the urban and rural participants were found for healthcare workers (85.5% vs. 65.5%), the Ministry of Health (70.5% vs. 52.6%; *p* = 0.036), the National Institute of Health (68.8% vs. 44%, *p* = 0.005), the WHO (67.7% vs. 48.3%; *p* = 0.043), the regional toll-free and emergency numbers (57.1% vs. 35.3%, *p* = 0.019), the provincial government (63.5% vs. 41.9%, *p* = 0.016), and the management of the South Tyrolean Health Service (25.8% vs. 46.6%). Trust in civil protection (63.3%) was not significantly different between the two groups.

Regarding information sources, trust in TV (33.3% of all participants (rather) trust), the press (32.8%), social media (11.5%), radio (37.5%), and famous people and influencers (10.2%) were not significantly different between urban and rural residents.

In the subgroup of parents of children aged 0–6, for TV (28.3%), press (30.7%), radio (38.8%), and influencers (9.5%), no significant difference in trust was found between urban and rural residents. There was a significant difference (*p* = 0.022) between the urban (17.7%) and rural (5.3%) participants only for trust in social media.

Conspiracy thinking—Of the five survey items on conspiracy theories for a total of 1426 urban and rural participants, the two statements “I think, that many very important things happen in the world, which the public is never informed about” (63.6% rural, 69.1% urban; *p* = 0.03) and “I think that events, which superficially seem to lack a connection are often the result of secret activities” (25.5% rural and 32.9% urban; *p* = 0.002) resulted as significantly more supported by urban residents. The statements ‘I think that the politicians usually do not tell us the true motives for their decisions’ (56.6% rural vs. 56.2% urban), “I think that government agencies closely monitor all citizens’ (27.8 rural vs. 29.1 urban), and “I think that there are secret organizations that greatly influence political decisions” (33.8% rural vs. 37.1% urban) did not show significant differences.

None of the five statements showed a significant difference between urban and rural residents in the subgroup of parents with children aged 0–6 years old.

Altruism, resilience, and well-being—Four of the five items on altruism were not significantly different for urban and rural residents when analyzing the frequencies of (rather) agree-answers of “I enjoy doing things for others” (79.9% rural vs. 80.7% urban), ‘I try to help others, even if they do not help me’ (75.9% rural vs. 76.3% urban), ‘Seeing others prosper makes me happy’ (87.2% rural vs. 84.5% urban) and ‘I care about the needs of other people’ (75.7% rural vs. 74.5% urban). Only the item ‘I come first and should not have to care so much for others’ (13.1% rural vs. 17.4% urban) found significantly more agreement in the urban population (*p* = 0.03). For the subgroup of parents of children aged 0–6 years, no significant differences were detected in questions regarding altruism.

The three items for resilience ‘I have a hard time making it through stressful events’ (27.1% rural vs. 34.2% urban (rather) agreed), ‘It does not take long to recover from a stressful event’ (50.8% rural vs. 54.1% urban) and ‘It is hard for me to fall asleep when something bad happens’ (32.9% rural vs. 36.3% urban) did not show significant differences between urban and rural participants for the total of 1426 as well as for the subgroup of children parents with 0–6 years old children. 

All five items on personal well-being within the last two weeks indicated significantly higher well-being in the rural population for the total of 1426 participants: 37.4% of the urban and 32.2% of the rural participants felt ‘happy and in a good mood’ (*p* = 0.041), 39.1% of the urban and 30.8% of the rural felt ‘calm and relaxed’ (*p* < 0.001), 46.5% of the urban and 37.5% of the rural felt ‘active and energetic’ (*p* < 0.001), 44.4% of the urban and 35.4% of the rural ‘woke up fresh and rested’ (*p* < 0.001) and 47.7% of the urban and 41.1% of the rural stated that their ‘everyday life was full of things that they interested’ (*p* = 0.019).

#### 3.3.3. Composite Scores for Trust in Information Sources and Institutions, Conspiracy Theories, Altruism, Resilience and Wellbeing

Composite (sum) scores were calculated for trust in information sources, institutions, conspiracy theories, altruism, resilience, and well-being. Significant differences between urban and rural residents were found in trust in institutions (*p* < 0.001), altruism (*p* = 0.046), resilience (*p* = 0.045), and well-being (*p* < 0.001). The total scores for trust in the media (*p* < 0.001), trust in institutions (*p* < 0.001), and conspiracy theories (*p* < 0.001) were significantly different between the vaccine-hesitant and non-hesitant participants (Figure 1).

### 3.4. Multivariate Logistic Regression for Predictors of Hesitancy

For the study sample, previous statistical analyses [21] identified age, chronic disease, high educational level, trust in institutions, conspiracy theories, trust in the national vaccination plan, and the frequency of information search as significant predictors of vaccine hesitancy. We added all additional predictors that were found to be significantly different between rural and urban participants and recalculated the regression model separately for the two groups.

All the models were tested for linearity of the logits of the metric predictors, including the corresponding quadratic terms. All quadratic terms were non-significant. Rural vaccine hesitancy was predicted by younger age (OR = 0.956; 95% CI, 0.937–0.975; *p* < 0.001), absence of chronic disease (OR = 0.230, CI = 0.067–0.792, *p* = 0.02), lower frequency of information searches (OR = 1.198, CI = 1.014–1.414, *p* = 0.033), higher resilience (OR = 1.154, CI = 1.019–1.308, *p* = 0.024), lower agreement with the national vaccination plan (OR = 0.187, CI = 0.128–0.271, *p* < 0.001), lower trust in institutions (OR = 0.900, CI = 0.871–0.930], *p* < 0.001), higher trust in conspiracy theories [OR = 1.061, CI = 1.016–1.114, *p* = 0.018), and higher altruism (OR = 1.080, CI = 1.016–1.149, *p* = 0.014) (Table 2).

Urban vaccine hesitancy was predicted by younger age (odds ratio [OR] = 0.954, 95% confidence interval [CI] = 0.954–0.999, *p* = 0.039), higher frequency of information searches (OR = 1.288, CI = 1.074–1.545, *p* = 0.006), lower agreement with the national vaccination plan (OR = 0.184, CI = 0.114–0.296, *p* < 0.001), lower trust in institutions (OR = 0.940, CI = 0.903–0.979], *p* = 0.003), and higher trust in conspiracy theories [OR = 1.164, CI = 1.085–1.249, *p* < 0.001). 

We found that chronic disease, altruism, and resilience were significant predictors in the rural group, but not in the urban group. Model diagnostics using DFBETA statistics showed that the model was stable and did not change after excluding single cases.

In a further logistic model for rural areas, single households and living with at-risk patients were included as independent predictors. None of the variables were statistically significant. The model is not presented because we have already found in [21] that neither variable is significant in an overall logistic regression model. In the logistic model for urban areas, “working in the health sector” was included as an independent predictor. It resulted in significance (*p* = 0.033), but at the same time, age was no more significant. Both variables are correlated since people over 64 no longer work in the health sector. Therefore, we included only age as a predictor.

## 4. Discussion

In our study, a systematic comparison of vaccination hesitancy among the South Tyrolean urban and rural populations revealed a more skeptical attitude toward COVID-19 vaccination in the rural population than in the urban population. The relative difference in quantitative terms was 37% and in absolute terms, 4.8% (from 12.8% in urban areas to 17.6% in rural areas). The demographic differences between the two population groups identified, in the rural sample, a lower level of education and a greater frequency of German as a mother tongue than Italian or Ladin, the three official languages of South Tyrol. People living in the countryside are less often chronically ill and are less affected by COVID-19 and death among family and friends. The economic impact of the pandemic affected the rural population more than the urban population. These findings suggest a continued need for COVID-19 vaccination interventions. Although age and gender did not differ significantly between people in urban and rural areas, more people in families with young children up to age 6 lived in rural areas. 

Health information seeking regarding COVID-19 and immunization-related information was lower in rural than in urban populations, as was trust in institutions and the national vaccination plan, and the absence of chronic diseases, while conspiracy thinking was more frequent in urban than in rural residents. Thus, most but not all known predictors of vaccine hesitancy are more common among rural than urban populations, which is consistent with lower vaccine uptake [6].

In the present study, altruism, and resilience, which are attitudes identified as predictors of vaccine uptake [15,35,36,37,38], showed significant differences between urban and rural residents. The level of individual resilience may correlate with the social resilience of both urban and rural people, and hesitant vaccination has been related to social resilience [39]. However, good individual resilience has also been associated with both increased and decreased vaccination readiness [38]. The role of altruism and resilience as predictors of vaccination hesitancy should be studied in more detail.

The urban population largely matched the rural population in the predictors of hesitant vaccination in multivariate logistic regression analyses. Chronic diseases, altruism, and resilience do not appear to play independent roles as risk predictors in urban populations. As it is difficult to influence all three variables through educational and other interventions to increase vaccination, there is no need for a fundamental intervention difference between urban and rural populations. 

COVID-19 vaccine hesitancy in South Tyrol in rural areas may not be a consequence of economic or other disparities but of a generally lower need to meet other people in sparsely populated areas. Others have found that sparse population density is an independent predictor of vaccine hesitancy [40]. Reservations regarding COVID-19 vaccination strongly depend on the sociocultural characteristics of the study population. In addition to knowledge about the safety and efficacy of vaccines, fear of side effects, and level of information about the disease and vaccination, religious attitudes may play an important role in rural areas [41]. Reduced interest in information and a greater lack of trust in institutions and information sources among rural residents were also confirmed. Thus, factors with a negative influence on vaccination behavior are more common in rural areas than in urban areas. This disparity may be important in that social determinants of coexistence are stronger in the countryside than in the city, and from social epidemiology, we know that their influence on attitudes and ideas, especially in relation to a pandemic, including vaccine hesitancy, is significant [2].

The negative pandemic experience and high pressure to be vaccinated to the point of mandatory COVID-19 vaccination with novel vaccines may have negatively influenced parents’ attitudes toward non-COVID-19 vaccination in young children. This pandemic may have made parents more reluctant to vaccinate their children against diseases caused by other viruses. Parental hesitation to vaccinate represents an additional barrier to compulsory vaccination and is likely to be partly responsible for the decline in children’s vaccinations observed during the pandemic [42]. In the present survey conducted in the second year of the pandemic, almost three-quarters of all participants (73.6%) reported that because of COVID-19 they had not changed their attitudes toward the compulsory vaccination of their children, which is valid in Italy. Nearly one-fifth (19.6%) of the participants reduced their parental hesitation to vaccinate their children. Only one in 20 participants developed a more negative attitude toward mandatory vaccination. No relevant differences in parental vaccine hesitancy were observed between urban and rural areas.

To promote willingness to be vaccinated, particularly among the rural population, it has been suggested that local and religious leaders, particularly community health workers or other health workers, overcome context-specific barriers in areas with low COVID-19 vaccine uptake [20]. The increasing shortage of healthcare personnel particularly affects primary care, especially general practitioners with German language skills, as working in the public health service in South Tyrol requires bilingualism (German and Italian), thus contributing to personnel shortages. Therefore, it could well be that the language group affiliation may lead to a selective lack of information with an impact on vaccination behavior. In this study, rural people more often reported German or Ladin as their mother tongue than Italian or other minority languages; however, neither for rural nor for urban people was language group affiliation a significant predictor of vaccine hesitancy.

In medical practice, we expect that trust in doctors and healthcare workers may be the starting point for improving trust in institutions and their decisions for the population to gain more confidence. High-quality, simple, and exhaustive information campaigns are necessary to reach the entire population because the frequency of information searches is a strong predictor of non-hesitancy. The information must be easily accessible and available in all languages. Regarding the differences in vaccine hesitancy between urban and rural areas, proposing specific recommendations for rural residents beyond the fundamental thematization of the identified predictors is more complex since, for example, chronic diseases, altruism, and resilience are factors that can hardly be influenced. 

Mistrust in medical institutions and health services by population segments has been studied extensively over the course of the COVID-19 pandemic, particularly in the context of the need for high vaccine uptake rates as part of pandemic management. Professional discussions on how to improve trust initially focused on the populations affected by mistrust. The burden of change has only subsequently changed from the affected populations to health institutions [43]. The present study describes the disparities in vaccination hesitancy between rural and urban areas in South Tyrol and opens up the possibility for institutions to specifically reduce mistrust. Particularly in the countryside, primary healthcare personnel can set confidence-building measures for effective information transfer, probably in the most effective manner [44].

The limitations of this study include unmeasured variables of potential importance for vaccine hesitancy, such as political and religious orientation or the use of alternative and complementary medicine, which are known predictors [45,46,47]. Furthermore, key sources of information on COVID-19 and vaccination were provided online, but habits related to Internet access and reading electronically versus printed sources of information were not surveyed, which may have limited the validity of the work. Another limitation is that the findings presented here on residence correlates of vaccine hesitancy from the second year of the pandemic are valid in later periods [44]. Finally, as described in [21], resilience was measured using only three of the six questions in a valid, reliable, and widely used questionnaire. This was reflected by a low Cronbach’s alpha value (0.66). Therefore, this measure should be cautiously interpreted. Finally, children and adolescents were excluded from the survey, leaving their impact on potentially important predictors of vaccine hesitancy in the context of rurality uninvestigated, although the vaccination of children and adolescents has become increasingly important during the pandemic.

## 5. Conclusions

Vaccine hesitancy is more frequent in rural populations than in urban populations, and residency is not an independent risk factor for vaccine hesitancy in economically well-developed South Tyrol. The urban population largely matched the rural population in terms of predictors of vaccine hesitancy. Chronic diseases, altruism, and resilience, which are modifying variables in the rural population, do not appear to play independent roles as risk predictors in the urban population. However, since all three variables are difficult to influence by educational and other interventions to increase vaccination, there is no need for a fundamental strategic difference in communications aimed at improving vaccination readiness between urban and rural populations. Nevertheless, noticeable demographic and individual disparities may inform public health measures targeting increased rural COVID-19 vaccine hesitancy. Larson et al. [44] in their review about vaccine hesitancy write: “Given that physicians and other healthcare providers are still among the most trusted when it comes to healthcare advice, local information about the nature and scope of vaccine hesitancy in their communities may help them anticipate and support important conversations in the clinic.” Thus, factors with greater prevalence in rural populations have been described and are amenable to targeted medical counseling, which may play a special role in interventions against vaccine hesitancy.

## Figures and Tables

**Figure 1 vaccines-10-01870-f001:**
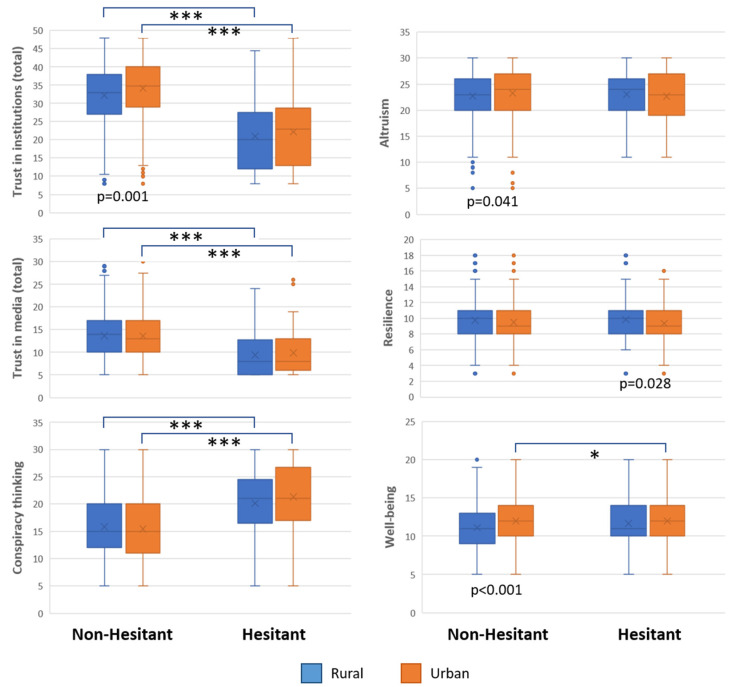
Box plots of (composite) scores for subgroups of rural/urban participants and answers to the question ‘Would you get vaccinated for COVID-19?” (vaccine hesitancy). *P*-values indicate differences between rural and urban residency; *, indicates differences between non-hesitant and hesitant. * *p* < 0.05, *** *p* < 0.001.

**Table 1 vaccines-10-01870-t001:** Characteristics of the sample and comparison between urban and rural residents.

Variable	Total *n* = 1426 N (%)	Rural *n* = 824 N (%)	Urban *n* = 602 N (%)	Urban vs. Rural ^†^
Age (years)				
18–34	334 (23.4)	203 (24.6)	131 (21.7)	n.s.
35–49	355 (24.9)	216 (26.2)	139 (23.1)	
50–64	391 (27.5)	215 (26.1)	176 (29.4)	
≥64	346 (24.2)	190 (23.1)	156 (25.9)	
Gender				
Male	691 (51.5)	397 (48.2)	294 (48.8)	n.s.
Female	735 (48.5)	427 (51.8)	308 (51.2)	
Education				
Middle school or lower	317 (22.2)	183 (22.2)	134 (22.2)	<0.001
Vocational school	411 (28.8)	288 (34.9)	123 (20.6)	
High school	411 (28.8)	234 (28.4)	177 (29.4)	
University	287 (20.1)	119 (14.5)	168 (27.9)	
Citizenship				
Italian	1308 (91.7)	763 (92.6)	545 (90.5)	n.s.
Other	118 (8.3)	61 (7.4)	57 (9.5)	
Native Language ^§^				
German	879 (61.7)	659 (80.0)	220 (36.6)	<0.001
Italian	384 (26.9)	70 (8.5)	314 (52.7)	
Ladin	57 (4)	50 (6.1)	7 (1.2)	
More than one/another language	106 (7.4)	45 (5.5)	61 (10.1)	
Household/Family structure (more than one answer possible)				
Single	236 (16.6)	137 (16.6)	99 (16.4)	n.s.
Children 0–6 years of age	178 (12.5)	116 (14.1)	62 (10.3)	0.034
Adolescents 7–17 years of age	277 (19.4)	174 (21.1)	103 (17.1)	n.s.
COVID-19 patients at risk ^§^	302 (21.2)	175 (21.2)	127 (21.1)	n.s.
None of the above	522 (36.6)	294 (35.7)	228 (37.9)	n.s.
Working in the health sector				
Yes	85 (6)	45 (5.5)	40 (6.7)	n.s.
No	1341 (94)	779 (85.0)	562 (79.6)	
Chronic disease(s)				
Yes	247 (17.3)	124 (15.0)	123 (20.4)	0.008
No	1179 (82.7)	700 (85.0)	479 (79.6)	
Relatives or friends died from COVID-19				
Yes	124 (8.7)	61 (7.4)	63 (10.5)	0.043
No	1302 (91.3)	763 (92.6)	539 (89.5)	
Economic situation (last 3 months)				
Better	43 (3)	20 (2.4)	23 (3.8)	0.003
The same	974 (68.3)	539 (65.5)	435 (72.2)	
Worse	375 (26.3)	246 (29.8)	129 (21.5)	
Don’t know	34 (2.4)	19 (2.3)	15 (2.5)	
Non-COVID-19 vaccine hesitancy				
Non-hesitant	157 (88.2)	99 (85.3)	58 (93.5)	n.s.
Hesitant	21 (11.8)	17 (14.7)	4 (6.5)	
COVID-19 vaccine hesitancy				
Non-hesitant	1204 (84.4)	679 (82.4)	525 (87.2)	---
Hesitant	222 (15.6)	145 (17.6)	77 (12.8)	

^†^*p*-values refer to chi-square tests for nominal and ordinal data and the Mann-Whitney U test for the metric variable age, comparing urban and rural participants. ^§^ Participants living with a COVID-19 patient at risk in a household. n.s., not significant.

**Table 2 vaccines-10-01870-t002:** Predictors of vaccination hesitancy in South Tyrol, Italy in March 2021 in multivariate logistic regression analysis.

	Rural *n* = 842 Nagelkerkes R^2^ = 0.627			Urban *n* = 602 Nagelkerkes R^2^ = 0.577		
	Regression Coefficient B	*p*-Value	OR = Exp(B) [95% CI]	Regression Coefficient B	*p*-Value	OR = Exp(B) [95% CI]
Age	−0.045	<0.0001	18.196 [0.937; 0.975]	−0.024	0.039	0.976 [0.954; 0.999]
High Educational level ^#^	−0.244	n.s.	0.784 [0.569; 1.080]	−0.252	n.s.	0.777 [0.549; 1.101]
Family and risk patterns						
Children 0-6 years of age	0.448	n.s.	1.565 [0.785; 3.121]	0.383	n.s.	1.466 [0.550; 3.909]
A family member or relative died from COVID-19	−0.532	n.s.	0.588 [0.200; 1.730]	−0.433	n.s.	0.648 [0.184; 2.282]
Chronic disease	−1.469	0.020	0.230 [0.067; 0.792]	−0.770	n.s.	0.463 [0.159; 1.344]
Search for information	0.180	0.033	1.198 [1.104; 1.414]	0.253	0.006	1.288 [1.074; 1.545]
Altruism (total score)	0.077	0.014	1.080 [1.016; 1.149]	0.009	n.s.	1.009 [0.939; 1.085]
Well-being (total score)	0.021	n.s.	1.021 [0.937; 1.113]	−0.022	n.s.	0.978 [0.879; 1.088]
Resilience (total score)	0.143	0.024	1.154 [1.019; 1.308]	−0.054	n.s.	0.948 [0.824; 1.090]
Trust						
Agree with the national vaccination	−1.679	<0.001	0.187 [0.128; 0.271]	−1.695	< 0.001	0.184 [0.114; 0.296]
Trust in institutions (total score)	−0.105	<0.001	0.900 [0.871; 0.930]	−0.062	0.003	0.940 [0.903; 0.979]
Mother tongue ^+^						
Italian		n.s.			n.s.	
German	−0.964	n.s.	0.381 [0.124; 1.168]	−0.392	n.s.	0.676 [0.304; 1.504]
Ladin	−0.744	n.s.	0.475 [0.111; 2.043]	−1.584	n.s.	0.205 [0.001; 68.762]
Another language	−0.227	n.s.	0.797 [0.191; 3.336]	0.109	n.s.	1.116 [0.425; 2.931]
Bad Economic situation	−0.057	n.s.	0.945 [0.637; 1.401]	0.130	n.s.	1.138 [0.425; 2.931]
Conspiracy thinking (total score)	0.059	0.018	1.061 [1.010; 1.114]	0.152	<0.001	1.164 [1.085; 1.249]

Constant term was not significant. *p*-values for the significant contribution of the independent variables to the model. ^#^ One possible answer is a categorical variable, with the first value used as an indicator. ^+^ Language is introduced as a dichotomous variable. n.s., not significant.

## Data Availability

The data presented in this study are available upon request from the corresponding author. The data were not publicly available because the survey was implemented by the statistical office of the regional administrative authority with politically sensitive content.

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
