# Peer review of "Rural-Urban Disparities in Vaccine Hesitancy among Adults in South Tyrol, Italy"

_vaccines, 2022, doi:10.3390/vaccines10111870_

Round 1
Reviewer 1 Report
Methods: Very good, detailed, also gives the correct definition of rurality. Among the analyzed aspects - which are correct and detailed - I missed however the access to internet, the habits regarding the relationship to printed and electronical press. At these are the most important sources of informations, the habits concerning the reading of printed and electronical press infuence basiclythe knowledge about COVID infection.
Results: Very good, scientifically based, informative. anyway a bit too long, but I have to remark that Table 1 and Figure 1 itself contain a plenty of very useful and interesting information.
Discussion: The real strength of the paper is this section. Wide-based, focuse and tries to discuss the very important sociological aspects in details. an interesting question: how to reduce the mistrust of people in institution. This is basic, not only concerning health care, but also otehr areas of the society. It would heve been interesting to compare the results to those of other researches.
Conclusion: very good
Author Response
|
Methods: Very good, detailed, also gives the correct definition of rurality. Among the analyzed aspects - which are correct and detailed - I missed however the access to internet, the habits regarding the relationship to printed and electronical press. At these are the most important sources of informations, the habits concerning the reading of printed and electronical press infuence basiclythe knowledge about COVID infection.
|
We thank the reviewer for his overall positive assessment of the methodology of our work.
Regarding the rightly missed information about access to the Internet, the lack of more detailed information on habits related to the use of printed and electronic press points to a weakness that has not yet been sufficiently discussed. Because this missing information has not been collected, we can only refer to it in the discussion as another limitation. We have added the following paragraph:
"Furthermore, key sources of information on COVID-19 and vaccination are provided online. Habits related to internet access and reading electronically versus printed sources of information were not surveyed, which may limit the validity of the work.“
|
|
Results: Very good, scientifically based, informative. anyway a bit too long, but I have to remark that Table 1 and Figure 1 itself contain a plenty of very useful and interesting information.
|
We thank the reviewer for the positive feedback.
To reduce the amount of data without loss of information, we reduced Table 1 by the column "Hesintant vs. Non-Hesitant." These data have been presented earlier and here the reference to the paper could be sufficient. The reduction of Table 1 is also suggested by Reviewer 2.
Following phrase was inserted: „Comparisons of demographic variables with respect to vaccine hesitancy were presented previously [21].“ |
|
Discussion: The real strength of the paper is this section. Wide-based, focuse and tries to discuss the very important sociological aspects in details. an interesting question: how to reduce the mistrust of people in institution. This is basic, not only concerning health care, but also otehr areas of the society. It would heve been interesting to compare the results to those of other researches.
|
We thank the reviewer for the positive feedback.
The following paragraph including an additional reference has been added tot he discussion section:
„Third, the mistrust in medical institutions and health services by segments of the population has been studied extensively over the course of the Covid 19 pandemic, particularly in the context of the need for high vaccine uptake rates as part of pandemic management. Professional discussions of how to improve trust have initially focused particularly on the populations affected by mistrust. The burden of change has only subsequently changed from affected populations to the health institutions [39]. The present study describes for South Tyrol the differences in vaccination hesitancy in rural and urban areas and opens the possibility for institutions to specifically reduce mistrust. Particularly in the countryside, primary health care personnel can set confidence-building measures for effective information transfer probably in the most effective manner [20].“
|
|
Conclusion: very good
|
We thank the reviewer for the positive feedback.
|
Reviewer 2 Report
The manuscript describe the determinats of vaccine hesitancy in rual Italy region.
I have the following commnets
1. The primary objective of the study is study teh determinants of Covid-19 vaccination in rural Italy but the results are primarily orineted towards hesistant versus non-hesitanata. In my opinion, results shall be oriented towards rural versus urban for hesistant and non-hesitant
2. Table 1: two p values are given for urban versus rural and hesitant versus non-hesitant which is causing confusion. They shall be separated.
3. Results shall be focussed towrads the primary objective
Author Response
|
The manuscript describe the determinats of vaccine hesitancy in rual Italy region. I have the following comments |
|
|
1. The primary objective of the study is study the determinants of Covid-19 vaccination in rural Italy but the results are primarily orineted towards hesistant versus non-hesitanata. In my opinion, results shall be oriented towards rural versus urban for hesistant and non-hesitant |
As also indicated in the title of the article, the focus of the study was on underreported vaccination hesitancy rather than actual vaccination rates in the northern-most Italian Province of Bolzano, which is located in the economically well-developted center of Europe. In fact, most results of the paper are oriented towards hesitancy vs. non-hesitancy. We have therefore inserted the following sentence in the abstract for clarification:
“Individual factors responsible for greater vaccination hesitancy in rural areas in the center of Europe are poorly understood and were therefore studied.“
|
|
2. Table 1: two p values are given for urban versus rural and hesitant versus non-hesitant which is causing confusion. They shall be separated. |
We thank the reviewer for the suggestion how to improve the data presentation. The table column with the second p-value has been removed and the following phrase and reference inserted in the text instead:
„Comparisons of demographic variables with respect to vaccine hesitancy were presented previously [21].“ |
|
3. Results shall be focussed towards the primary objective |
The focus on vaccine hesitancy was increased in the revision of the manuscript by adding an additional table with corresponding data description in the text:
Table S1 entiteld „Baseline characteristics for differences between urban and rural residents for the groups of hesitant and non-hesitant participants” and corresponding text in main manuscript inserted.
An additional paragraph was inserted on logistic regression analysis:
„In a further logistic model for the rural areas, single household and living together with patients at risk were included as independent predictors. Both variables resulted to be not significant. The model is not presented, since we have already found in [21], that both variables were not significant in an overall logistic regression model. In a further logistic model for urban areas, “working in the health sector” was included as an independent predictor. It resulted significant (p=0.033), but at the same time, age was no more significant. Both variables are correlated, since people over 64 do no more work in the health sector. Thus, we decided, to only include age as predictor.“
|
Round 2
Reviewer 2 Report
The authors have responded to my commnets adequately
Author Response
As reviewer 2 commented that "the authors have responded to my commnets adequately", no additional reply appears necessary.